# Atrial Fibrillation Detection Based on a Residual CNN Using BCG Signals

Qiushi Su [1], Yanqi Huang [1,*], Xiaomei Wu [1,2,3,4,5,*], Biyong Zhang [6,7], Peilin Lu [8] and Tan Lyu [9]

1   Center for Biomedical Engineering, School of Information Science and Technology, Fudan University, Shanghai 200438, China
2   Academy for Engineering and Technology, Fudan University, Shanghai 200433, China
3   The Key Laboratory of Medical Imaging Computing and Computer Assisted Intervention (MICCAI) of Shanghai, Shanghai 200032, China
4   Yiwu Research Institute, Fudan University, Yiwu 322000, China
5   Research Center of Assistive Devices, Shanghai 200093, China
6   Eindhoven University of Technology, Eindhoven 5612AP, The Netherlands
7   BOBO Technology, Hangzhou 311000, China
8   Department of Neurology, Neuroscience Center, Sir Run Run Shaw Hospital, School of Medicine, Zhejiang University, Hangzhou 310016, China
9   Department of Electrocardiography, Sir Run Run Shaw Hospital, School of Medicine, Zhejiang University, Hangzhou 310016, China
*   Correspondence: yqhuang@fudan.edu.cn (Y.H.); xiaomeiwu@fudan.edu.cn (X.W.)

**Abstract:** Atrial fibrillation (AF) is the most common arrhythmia and can seriously threaten patient health. Research on AF detection carries important clinical significance. This manuscript proposes an AF detection method based on ballistocardiogram (BCG) signals collected by a noncontact sensor. We first constructed a BCG signal dataset consisting of 28,214 ten-second nonoverlapping segments collected from 45 inpatients during overnight sleep, including 9438 for AF, 9570 for sinus rhythm (SR), and 9206 for motion artifacts (MA). Then, we designed a residual convolutional neural network (CNN) for AF detection. The network has four modules, namely a downsampling convolutional module, a local feature learning module, a global feature learning module, and a classification module, and it extracts local and global features from BCG signals for AF detection. The model achieved precision, sensitivity, specificity, F1 score, and accuracy of 96.8%, 93.7%, 98.4%, 95.2%, and 96.8%, respectively. The results indicate that the AF detection method proposed in this manuscript could serve as a basis for long-term screening of AF at home based on BCG signal acquisition.

**Keywords:** ballistocardiogram; atrial fibrillation; residual neural network; classification

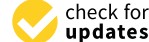



## 1. Introduction

Atrial fibrillation (AF) is the most common clinical arrhythmia, threatening patient health and profoundly increasing morbidity, mortality, and healthcare-related costs [1]. At present, the estimated prevalence of AF in adults is between 2% and 4%, with a 2.3-fold increase projected due to increased lifespan in the general population [2]. The risk of developing AF increases with age. Approximately 60% of patients with AF are between the ages of 65 and 85 years [3]. Various diseases have been associated with AF, such as stroke, heart failure, coronary artery disease, and systemic thromboembolism [4,5]. Thus, the timely detection and prevention of AF is significant [6]. However, a significant proportion of patients with atrial fibrillation are asymptomatic or have mild symptoms, a phenomenon known as subclinical atrial fibrillation. Thus, without the use of cardiac monitoring devices, timely and accurate diagnoses are difficult. Therefore, a cardiac monitoring device that can be used for daily monitoring and early detection of AF is needed.

At present, electrocardiography (ECG) is the main method for diagnosing AF. The typical features of AF include the absence of P-waves, which are replaced by rapid oscillations or fibrillatory waves, and the irregular variability of RR intervals. These features have allowed various AF detection methods to be developed, including the template-based algorithm [7], feature analysis based on the RR intervals [8], the application of symbolic dynamics, Shannon entropy based on the support vector machine (SVM) [9], and the combination of the genetic algorithm and random forest classifier [10]. In recent years, convolutional neural networks (CNNs) have been used for AF detection due to their strong feature extraction ability. Limam et al. [11] proposed a convolutional recurrent neural network (CRNN) consisting of two independent CNNs to extract related features, one from ECG and the other from heart rates. Dang et al. [12] combined CNN and bidirectional long short-term memory (Bi-LSTM) to automatically detect AF from ECG signals. Yao et al. [13] proposed a multi-scale CNN which applied time scaling on input ECG signals and detected AF based on scaled inputs. The residual network [14] is a type of deep neural network (DNN) that was first proposed for image classification tasks. The residual network has recently been successfully applied to ECG AF detection due to its excellent classification performance and characteristics that can alleviate the degradation problem of DNNs. He et al. [15] proposed a model consisting of residual CNN and Bi-LSTM to extract features from raw ECG signals. Cao et al. [16] used an improved multi-scale decomposition enhanced residual CNN to detect AF from a single short ECG lead recording. Faust et al. [17] proposed a method using residual CNN model to extract features from RR intervals of ECG signals. However, the above methods are based on the acquisition of ECG signals and usually require that electrodes are in direct contact with the skin of the subject, which can be uncomfortable and inconvenient for long-term use, and is thus not suitable for monitoring cardiac activity in daily life. Therefore, researchers have proposed some noncontact cardiac activity monitoring techniques.

Previous studies have shown that noncontact cardiac monitoring techniques can be used by subjects in daily life for long-term cardiac activity monitoring [18]. The ballistocardiogram (BCG) [19] and seismocardiogram (SCG) [20] capture the body's mechanical responses to cardiac activity and blood circulation. The difference between them is that the BCG measures pressure changes of the body against the measurement plane, while the SCG measures local vibrations of the chest wall. The BCG is a noncontact body vibration detection method. In addition to the mechanical activity of the heart, other physiological factors that cause body vibrations include breathing, noise, and motion artifacts (MA) [21–23]. BCG signals can be acquired by placing piezoelectric sensors on the measurement plane, such as under mattresses [24] and chairs cushions [25]. In recent years, researchers have proposed many cardiac activity detection algorithms based on BCG signals, such as heart rate detection [26,27], J peak feature extraction [28], and cardiovascular disease classification [29].

Although BCG-based cardiac activity monitoring research has developed rapidly in recent years, the use of BCG signals for AF detection remains an unexplored technique. Brüser et al. [30] proposed a feature selection algorithm based on the mutual information between the features and class labels, as well as the first- and second-order interactions among features, and evaluated seven machine learning algorithms (naive Bayes, linear and quadratic discriminant analysis, support vector machine, random forest, and bagged and boosted trees) for their performance in separating 30 s-long BCG epochs into AF, SR, and MA. The best classifier (random forest) achieved a mean sensitivity and specificity of 93.8% and 98.2%, respectively. Yu et al. [31] extracted features from stationary wavelet transform of 30 s BCG epochs and used three machine learning classifiers (support vector machine, K-nearest neighbor, and ensemble learning) to detect AF. The ensemble classifier achieved a mean accuracy, sensitivity, and specificity of 94.4%, 97.0%, and 89.1%, respectively. Wen et al. [32] extracted features from BCG energy signals and used five machine learning algorithms (support vector machine, naive Bayes, decision tree, bootstrap aggregated decision tree, and random forest) to identify AF and SR. The algorithm achieved a

sensitivity, precision, and accuracy of 96.8%, 92.8%, and 94.5%, respectively. Jiang et al. [33] proposed a deep learning method for AF and SR classification that integrated features extracted from a Bi-LSTM network and features extracted from phase space. The method achieved an accuracy, specificity, sensitivity, and precision of 94.7%, 93.5%, 95.9%, and 93.7%, respectively.

Some effective methods for BCG-based AF detection have been proposed thus far, but certain issues remain. First, most previous studies [30–32] have focused on traditional machine learning. However, manual feature extraction is subjective and may lead to the loss of important information, especially considering the morphological diversity of BCG signals. Second, the BCG datasets used with the previously proposed methods [30–33] are small in scale, while machine learning, especially deep learning methods, require a large amount of data for training and validation.

In this manuscript, we focus on developing a noncontact AF detection method based on BCG signals. An offline residual CNN model, inspired by ECG-based AF detection methods [15–17], was proposed to detect AF in BCG segments, which can be applied in household long-term AF monitoring and screening. To summarize, the main contributions are: (1) To the best of our knowledge, this manuscript is the first to apply residual CNN to extract features from BCG segments for AF detection. (2) The number of BCG segments in our dataset is currently the largest in relevant research, which ensures the feasibility and reliability of the proposed deep learning method. This manuscript is organized as follows: Section 2 describes the proposed method including dataset construction, residual CNN model construction, and the detailed experimental process. Section 3 presents our study results and analysis. Section 4 provides discussions of our findings and potential future work. Section 5 concludes the whole passage.

## 2. Materials and Methods

In the proposed method, the BCG signals were collected from 45 inpatients by a non-contact BCG acquisition system. The discrete wavelet transform and RMS filter were then used to preprocess the acquired BCG signals. After that, we labeled the BCG recordings according to the synchronously collected ECG signals and split them into 10 s nonoverlapping segments consisting of three types of BCG signals (AF, SR, and MA). A residual CNN model was introduced to classify the classification task for AF detection. The overall flow diagram of the proposed method is shown in Figure 1.

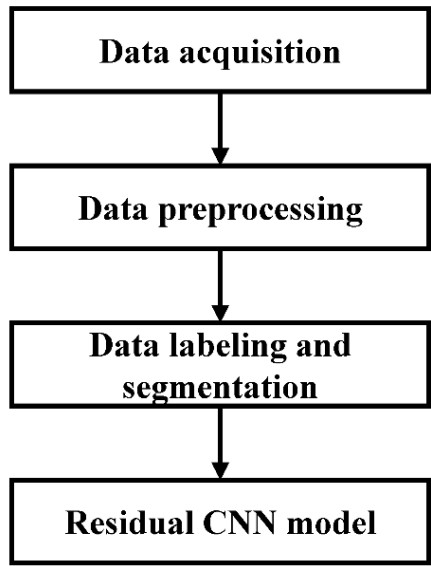

**Figure 1.** Flow diagram of the proposed method.

## 2.1. Data Acquisition

### 2.1.1. Participants

In this study, we collected BCG signals (509.8 h in total) from 45 subjects (24 males and 21 females, aged 27–93 years, mean 72.6 years (standard deviation (STD) 14.8), who were hospitalized in Sir Run Run Shaw Hospital, School of Medicine, Zhejiang University, Hangzhou, Zhejiang, China. Among these subjects, 19 had confirmed paroxysmal AF, and 26 had non-AF. Detailed demographic information of these subjects is presented in Table 1. The BCG data collection was approved by the Ethics Committee of Sir Run Run Shaw Hospital, School of Medicine, Zhejiang University, and the subjects.

**Table 1.** The demographic information of the subjects.

|  | AF | Non-AF | ALL |
|---|---|---|---|
| Amount | 19 | 26 | 45 |
| Gender (male/female) | 8/11 | 16/10 | 24/21 |
| Age (years) | $74.3 \pm 10.0$ | $71.3 \pm 17.4$ | $72.6 \pm 14.8$ |
| BMI (kg/m²) | $21.5 \pm 3.3$ | $20.9 \pm 2.9$ | $21.2 \pm 3.1$ |
| Average acquisition duration (hours) | $12.2 \pm 3.4$ | $10.7 \pm 1.6$ | $11.3 \pm 2.6$ |

### 2.1.2. BCG Recording

In this study, a noncontact BCG signal acquisition system (Figure 2a) designed and developed by Hangzhou BOBO Technology Ltd., Hangzhou, China, was used. The system consisted of a microcontroller, a signal conditioning circuit including amplifier, filter and analog-to-digital converter, and a piezoelectric sensor stripe with 7 cm width and 72 cm length. The sensitivity of the sensor was $0.015 \, \text{V}/10^{-6}\varepsilon$, where $\varepsilon$ is a dimensional value that represents the relative change of the length of a material to its initial length. The designed circuit included a Digital Automatic Gain Control System to guarantee the signal captured could almost always fit our requirements. Four sets of such a BCG signal acquisition system were used in this study, and these systems had the same specifications.

The sensor stripe was placed above the mattress but under the sheet of the hospital bed, which ensured no stress to subjects during sleep. It was horizontally placed under the subject's cardiac location to capture the best signal as shown in Figure 2b. BCG signal data were continuously collected while each subject slept on a hospital bed for the entire night without constraints. This means that the subjects could change their sleeping postures freely and might get out of bed to urinate, for instance. The BCG signals were recorded at 125 Hz sampling rate. Simultaneously, the ECG signals were collected by Holter (Beneware CT-08S) at a sampling rate of 200 Hz for reference.

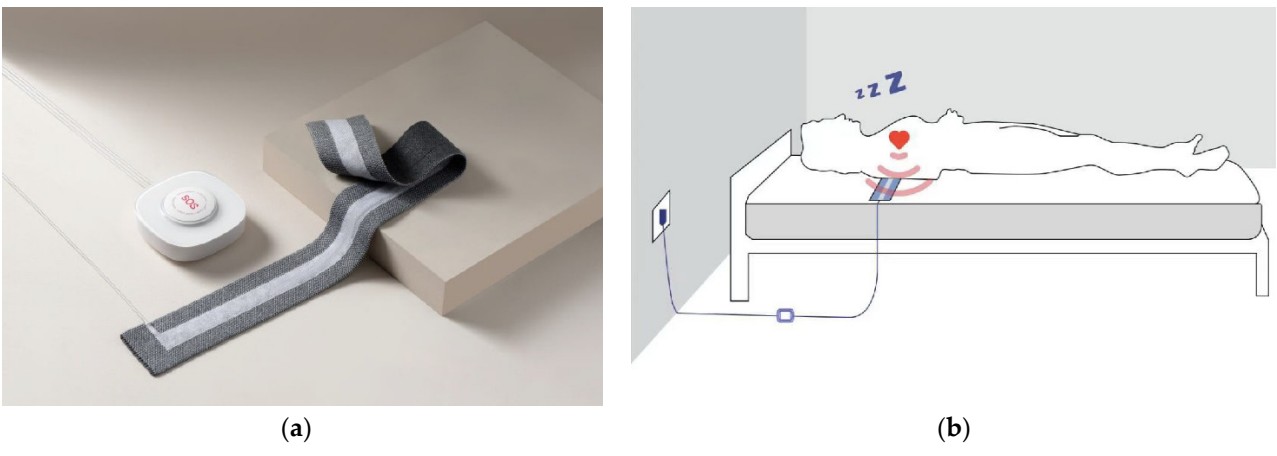

(**a**)    (**b**)

**Figure 2.** (**a**) The BCG signal acquisition system. (**b**) Schematic diagram of BCG acquisition.

## 2.2. Data Preprocessing

As mentioned in the introduction, in addition to heartbeat information, the raw BCG signals contain noise and interference from various sources, including motion artifacts due to the body movement of the subject. These interference, noise, and motion signals can seriously affect AF detection in BCG signals. Therefore, the collected BCG signals must be preprocessed to remove the interference and noise and separate the motion artifacts for use in the classification algorithm. In this manuscript, we used the same discrete wavelet transform and RMS filter as in a previous study by our group [32] to process the raw BCG signals. The acquired BCG signals were first decomposed into seven details, and then the third to sixth layers of details were selected for reconstruction.

The RMS filter, defined in Equation (1), was used to detect both motion artifacts during acquisition and invalid signals when the subject left their bed.

$$S[n] = \sqrt{\frac{1}{N}\sum_{k=0}^{N-1}(f[n+k])^2} \tag{1}$$

where $f[n]$ and $S[n]$ represent the BCG signal after the wavelet transform and the RMS-filtered BCG signal, respectively. $n$ represents the sample points, and $N$ represents the window size for calculating the root mean square value of each sample point, which was empirically set to 200. To unify the selection criteria of different subjects, we normalized the amplitude of $S[n]$ to [0, 1], that is, divided the amplitude by the maximum. The upper and lower thresholds of valid signals were set to 0.25 and 0.1, respectively, on the basis of empirical values (Figure 3). The segments with amplitudes less than 0.1 were discarded as invalid signals caused by the subject's deviation or leaving from the sensor (6.2 h in total). The segments with the amplitude higher than 0.25 were labeled motion artifacts and were reserved to build the BCG dataset as one of the three signal types. The rest of the BCG signals awaited further processing.

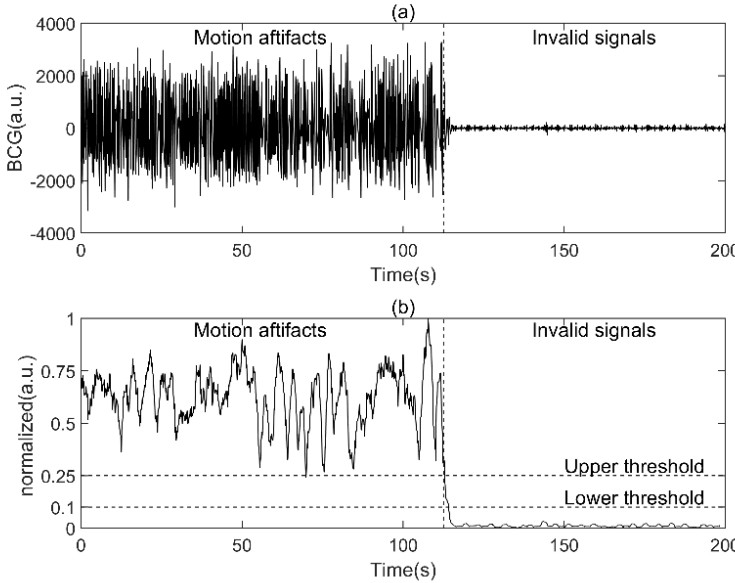

**Figure 3.** Detection of motion artifacts and invalid signals using RMS filter. (a.u.: arbitrary units): (**a**) raw BCG signal, (**b**), the corresponding normalized RMS−filtered BCG signal.

## 2.3. Data Labeling and Segmentation

In this section, the BCG signals were labeled according to the synchronously collected ECG and confirmed by experts (examples are shown in Figure 4). BCG signals from non-AF subjects were labeled SR if a normal sinus rhythm was confirmed by the synchronized ECG. BCG signals from AF subjects were labeled AF if AF rhythm was confirmed by

the synchronized ECG. Then, we divided the two types of labeled BCG signals and MA BCG signals detected in the signal preprocessing into 10 s nonoverlapping segments. All other BCG signals (425.2 h in total) were discarded during the labeling and segmentation process. Finally, the BCG dataset was constructed (78.4 h in total), as shown in Table 2. Mathematically, the dataset can be described as a $m \times n$ matrix, where $m$ represents the total number of BCG segments and $n$ represents the total number of sampling points for each segment, which equal 28,214 and 1250, respectively.

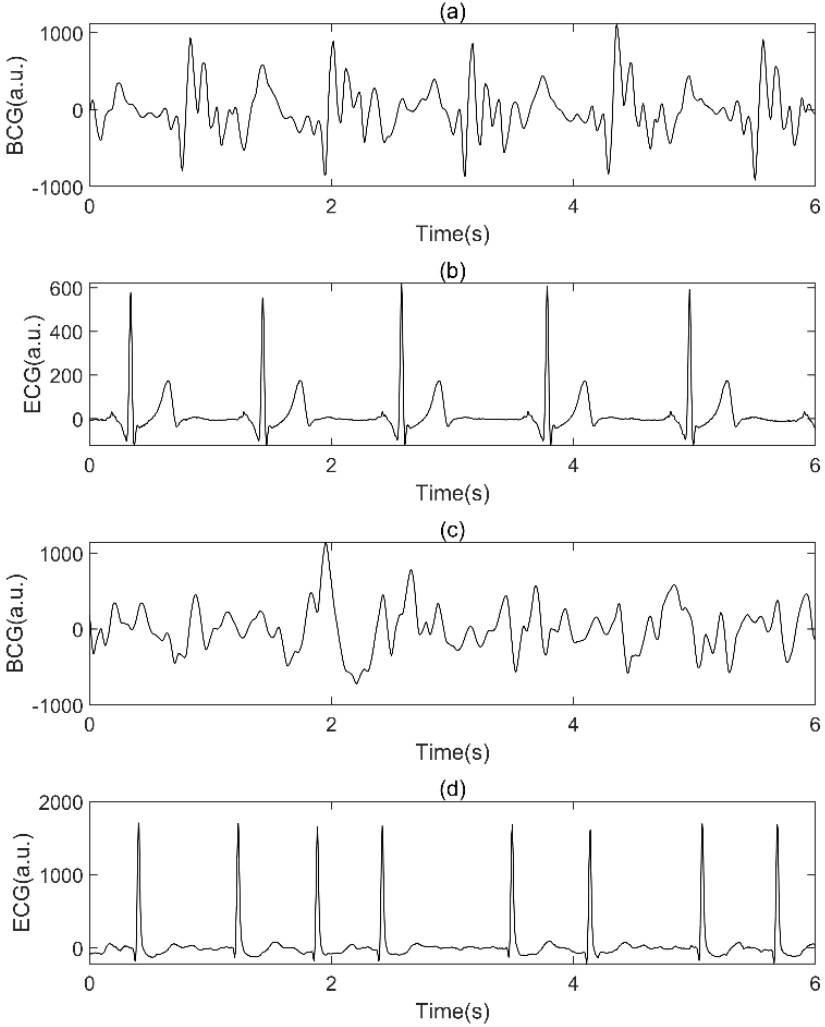

**Figure 4.** Examples of BCG signal and synchronized ECG. (a.u.: arbitrary units): (**a**) SR BCG signal, (**b**) synchronized ECG of (**a**), (**c**) AF BCG signal, (**d**): synchronized ECG of (**c**).

**Table 2.** The BCG dataset used in this research.

|  | AF | SR | MA | ALL |
|---|---|---|---|---|
| Amount of segments | 9438 | 9570 | 9206 | 28,214 |

Figure 5 shows the three types of BCG signals from different subjects in the BCG dataset. The amplitudes of MA are generally larger than the two other classes among subjects. The morphology of the BCG signals of AF and SR varied greatly among subjects due to differences in the physical condition, age, and sleeping posture of the patients, as well as the relative positions of the body and sensor caused by difference sleeping posture of the subjects, which increased the difficulty of screening for AF.

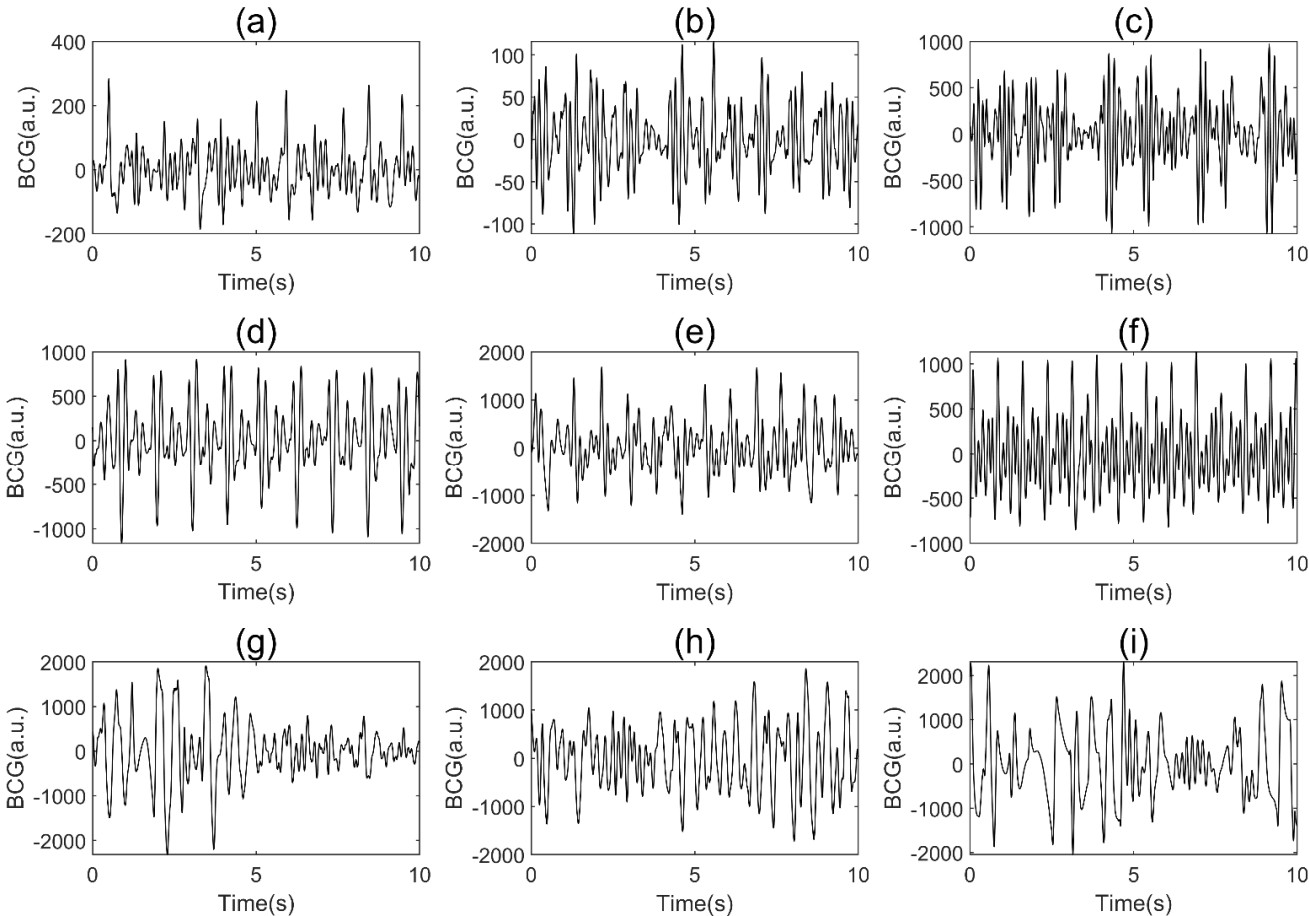

**Figure 5.** Examples of BCG signals from different subjects in our BCG dataset (a.u.: arbitrary units): (**a–c**), AF; (**d–f**), SR; (**g–i**), MA.

### 2.4. Neural Network Model Construction

#### 2.4.1. Problem Formulation

The AF detection task in this manuscript is to identify AF signals in single-channel BCG signals. The training set $T = \{(x^{(1)}, y^{(1)}), (x^{(2)}, y^{(2)}), \ldots, (x^{(i)}, y^{(i)}), \ldots, (x^{(k)}, y^{(k)})\}$ consists of the input BCG signal $x^{(i)}$ and label $y^{(i)}$, where $x^{(i)} \in R^n$ and $y^{(i)} \in \{0, 1, 2\}$. Here $k$ represents the total number of BCG segments in the training set. The labels $y^{(i)} = 0$, $y^{(i)} = 1$ and $y^{(i)} = 2$ correspond to AF, SR and MA, respectively. The residual CNN model constructed in this manuscript accepts $x^{(i)}$ as input and $\hat{y}^{(i)}$ as output, as shown in Equation (2):

$$\hat{y}^{(i)} = F(x^{(i)}; \theta) \tag{2}$$

where $F(\cdot)$ is the function of the model designed in this manuscript, $\theta$ are the related parameters, and $\hat{y}^{(i)}$ is the predicted label of our model, with $\hat{y}^{(i)} \in \{0, 1, 2\}$.

#### 2.4.2. Model Architecture

BCG signals, similar to ECG signals, are time series signals that reflect cardiac activity. The J wave in the BCG signal is similar to the R wave in the ECG signal, indicating that the BCG signal has a certain periodicity. However, BCG signals have considerable individual differences due to the physical conditions of the subjects and substantial variability due to differences in the measurement environment. Thus, inspired by ECG-based AF detection, a CNN with residual architecture consisting of 12 layers for detecting AF in BCG signals by extracting local and global features was developed. As shown in Figure 6, this proposed network has four modules, namely a downsampling convolutional module, a local feature

learning module, a global feature learning module, and a classification module. The detailed parameter configurations of the model are outlined in Table 3 (k, s, and *p* represent kernel size, stride, and padding respectively, Conv-n represents the number of channels n).

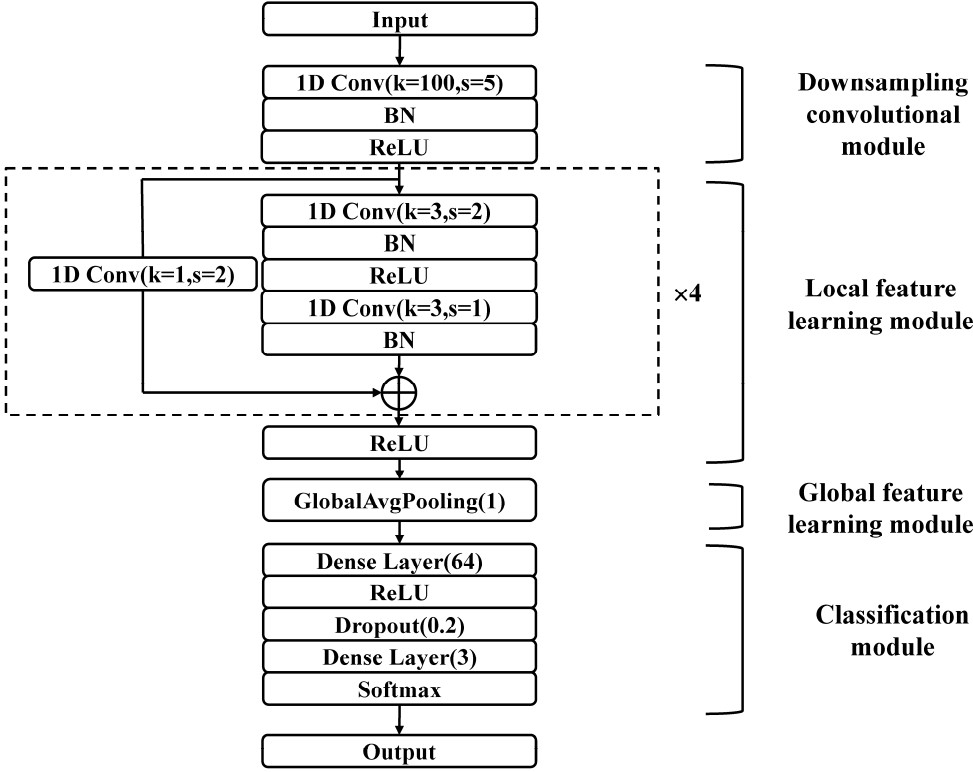

**Figure 6.** The structure of the proposed residual CNN for AF detection.

**Table 3.** The detailed parameters of the proposed residual CNN.

| Layer Name | Parameter Value |
| --- | --- |
| Layer 1 | Conv-16, k = 100, s = 5, $p = 0$ |
| Layer 2 | Conv-32, k = 3, s = 2, $p = 1$ |
| Layer 3 | Conv-32, k = 3, s = 1, $p = 1$ |
| Layer 4 | Conv-64, k = 3, s = 2, $p = 1$ |
| Layer 5 | Conv-64, k = 3, s = 1, $p = 1$ |
| Layer 6 | Conv-128, k = 3, s = 2, $p = 1$ |
| Layer 7 | Conv-128, k = 3, s = 1, $p = 1$ |
| Layer 8 | Conv-256, k = 3, s = 2, $p = 1$ |
| Layer 9 | Conv-256, k = 3, s = 1, $p = 1$ |
| Layer 10 | GlobalAvgPooling(1) |
| Layer 11 | Dense(64) |
| Layer 12 | Dense(3) |

The input of the model is the preprocessed BCG signal, which is a three-dimensional matrix with dimensions of (batch size, 1, 1250). The batch size is determined through experiments. The other two dimensions are the channel number and signal length. The signal first enters the downsampling convolutional module, which consists of a one-dimensional convolutional (1D Conv) layer, a batch normalization (BN) [34] layer, and a rectified linear unit (ReLU) [35] activation layer. The downsampling convolutional module quickly compresses the long BCG signal into a considerably shorter series of feature vectors. This module was designed to capture the contextual information of a BCG signal over a relatively long window based on the fact that the BCG signal is approximately periodic. Furthermore, this

module facilitates model training by preventing the stacking of too many convolutional layers.

In the local feature learning module, four stacked residual convolutional blocks are utilized to extract local features from the output of the downsampling convolutional module. Each residual block consists of two 1D Conv layers, two BN layers, two ReLU activation layers, and one shortcut connection. The feature vectors are halved as they pass through a residual block, and the number of channels is doubled. A 1D Conv layer with a convolution kernel of 1 and a stride of 2 is applied in the shortcut connection to match the channel number of the input and output.

Then, the extracted feature vectors are input into a global average pooling (Global-AvgPooling) [36] layer to learn the global features. This module reduces the number of parameters, preventing overfitting. According to the extracted global features, the classification module divides the BCG signals into different classes. The classification module consists of two dense layers, a ReLU activation layer, a dropout layer [37], and a softmax layer. The first dense layer has 64 cells, while the second layer has 3 cells, corresponding to the 3 classes. The dropout layer prevents overfitting by randomly discarding some of the neurons, improving the generalizability of the model. Finally, the corresponding classification probability is calculated by the softmax layer, as shown in Equation (3):

$$p(z^{(i)}) = \frac{\exp(z^{(i)})}{\sum_j \exp(z^{(i)})} \tag{3}$$

where $z^{(i)}$ is the output value of the last dense layer, and $p(\cdot)$ is the probability of the *i*-th output label having an input value of $z^{(i)}$. To evaluate the performance of the model on the training data, we use the cross-entropy loss, which is suitable for classification problems. For a training set containing *k* BCG segments, the loss function can be defined as shown in Equation (4):

$$L(X) = -\frac{1}{k} \sum_{i=1}^{k} \sum_{j=0}^{2} I\{y^{(i)} = j\} \log p(z^{(i)}) \tag{4}$$

where $I\{\cdot\}$ is the indicator function. The remaining parameters are described in detail in the Problem Formulation section.

### 2.5. Experimental Setup and Evaluation Metrics

2.5.1. Experimental Environment and Procedure

The neural network model proposed in this manuscript was developed by Python 3.8, and trained and tested on the deep learning framework PyTorch 1.9.1 on an Ubuntu 20.04 operating system. The model training and testing were performed on a computing server equipped with 4 NVIDIA RTX 3090 graphics cards with 24 GB memory each.

The experimental process included two steps. First, the three types of BCG signal segments were divided randomly into a training set, validation set and test set at a ratio of 7:1:2. The model was first evaluated with different numbers of residual blocks from multiple perspectives to determine the final structure. Then, tenfold cross validation was used to further evaluate the stability of the model. We randomly partitioned BCG segments into 10 equal-sized complementary subsets. Of the 10 subsets, a single subset was retained as the validation data for testing the model, and the remaining 9 subsets were used as training data. The cross-validation process was then repeated 10 times, with each of the 10 subsets used exactly once as the validation data.

During the whole experiment, we set the number of epochs (the number of times the entire BCG training set is trained) to 50 and the batch size (the number of BCG segments used for each batch of model training) to 32. The network parameters were updated using mini-batch gradient descent [38], and the Adam algorithm [39] was used to optimize the network. The learning rate (initial value is $2 \times 10^{-5}$) in the Adam algorithm uses a piecewise function decay, that is, the decay is 0.1 times the original every 10 epochs.

### 2.5.2. Evaluation Metrics

The confusion matrix of a certain class $j(j \in \{0, 1, 2\})$ in this manuscript is shown in Table 4, where $\hat{y}$ is the predicted label.

**Table 4.** Confusion matrix.

| | | **Predicted Condition** | |
| | | $y = j$ | $y \neq j$ |
|---|---|---|---|
| **True Condition** | $\hat{y} = j$ | True positive (TP) | False negative (FN) |
| | $\hat{y} \neq j$ | False positive (FP) | True negative (TN) |

To evaluate the classification effect of the model from multiple perspectives, the precision (Pre), sensitivity (Sen), specification (Spe), F1 score (F1), and accuracy (Acc) [40] were used as evaluation metrics in this manuscript. On the basis of Table 4, the evaluation metrics can be defined as shown in Equations (5)–(9):

$$Pre = \frac{TP}{TP + FP} \tag{5}$$

$$Sen = \frac{TP}{TP + FN} \tag{6}$$

$$Spe = \frac{TN}{TN + FP} \tag{7}$$

$$F1 = 2\frac{Pre \times Sen}{Pre + Sen} \tag{8}$$

$$Acc = \frac{1}{k} \sum_{i=1}^{k} I\{y^{(i)} = \hat{y}^{(i)}\} \tag{9}$$

## 3. Results

The effect of the number of residual blocks on the experimental results was explored to determine the final model structure first (Table 5).

**Table 5.** The impact of the number of residual blocks.

| Number of Residual Blocks | | 2 | 3 | 4 | 5 | 6 |
|---|---|---|---|---|---|---|
| Pre | AF | 93.3% | **96.3%** | 95.3% | 95.6% | 96.3% |
| | SR | 93.4% | 92.1% | **94.6%** | 92.4% | 92.1% |
| | MA | **100.0%** | 100.0% | 100.0% | 100.0% | 100.0% |
| | mean | 95.6% | 96.1% | **96.6%** | 96.0% | 96.1% |
| Sen | AF | 93.3% | 91.6% | **94.5%** | 91.9% | 91.6% |
| | SR | 93.4% | **96.6%** | 95.4% | 95.9% | 96.6% |
| | MA | **100.0%** | 100.0% | 100.0% | 100.0% | 100.0% |
| | mean | 95.6% | 96.1% | **96.6%** | 95.9% | 96.1% |
| Spe | AF | 96.6% | **98.5%** | 97.7% | 97.9% | 98.2% |
| | SR | 96.6% | 95.6% | **97.2%** | 95.9% | 95.7% |
| | MA | **100.0%** | 100.0% | 100.0% | 100.0% | 100.0% |
| | mean | 97.7% | 98.0% | **98.3%** | 97.9% | 98.0% |
| F1 | AF | 93.3% | 93.9% | **94.9%** | 93.8% | 93.9% |
| | SR | 93.4% | 94.4% | **95.0%** | 94.1% | 94.3% |
| | MA | **100.0%** | 100.0% | 100.0% | 100.0% | 100.0% |
| | mean | 95.6% | 96.1% | **96.6%** | 96.0% | 96.1% |
| Acc | | 95.5% | 96.1% | **96.6%** | 95.9% | 96.0% |

For models with number of residual blocks ranging from 2 to 6, the performance for each class, as well as the mean performance over all classes, are given. We highlighted the best results for each performance measure. When the number of residual blocks was set to 4, most of the metrics achieved the best performance. In particular, the sensitivity of AF is significantly better than others, which is an important screening metric because it measures the ability of the method to correctly detect AF.

To further evaluate the generalizability and stability of the model, we used tenfold cross validation when the number of residual blocks was set to 4. The final cross validation confusion matrix (the sum of the results of 10 computational experiments) and performance evaluation are shown in Tables 6 and 7. The ROC curves shown in Figure 7 provides a graphical representation of the classification results of all 10 folds.

**Table 6.** Confusion matrix of tenfold cross validation.

| True Condition | Predicted Condition | | |
| --- | --- | --- | --- |
| | **AF** | **SR** | **MA** |
| AF | 8844 | 594 | 0 |
| SR | 297 | 9273 | 0 |
| MA | 0 | 0 | 9206 |

**Table 7.** Performance evaluation (mean ± STD).

| | **AF** | **SR** | **MA** |
| --- | --- | --- | --- |
| Pre | 96.8% ± 0.8% | 94.0% ± 1.4% | 100.0% ± 0.0% |
| Sen | 93.7% ± 1.6% | 96.9% ± 0.8% | 100.0% ± 0.0% |
| Spe | 98.4% ± 0.4% | 96.8% ± 0.8% | 100.0% ± 0.0% |
| F1 | 95.2% ± 0.8% | 95.4% ± 0.7% | 100.0% ± 0.0% |
| Acc | | 96.8% ± 0.5% | |

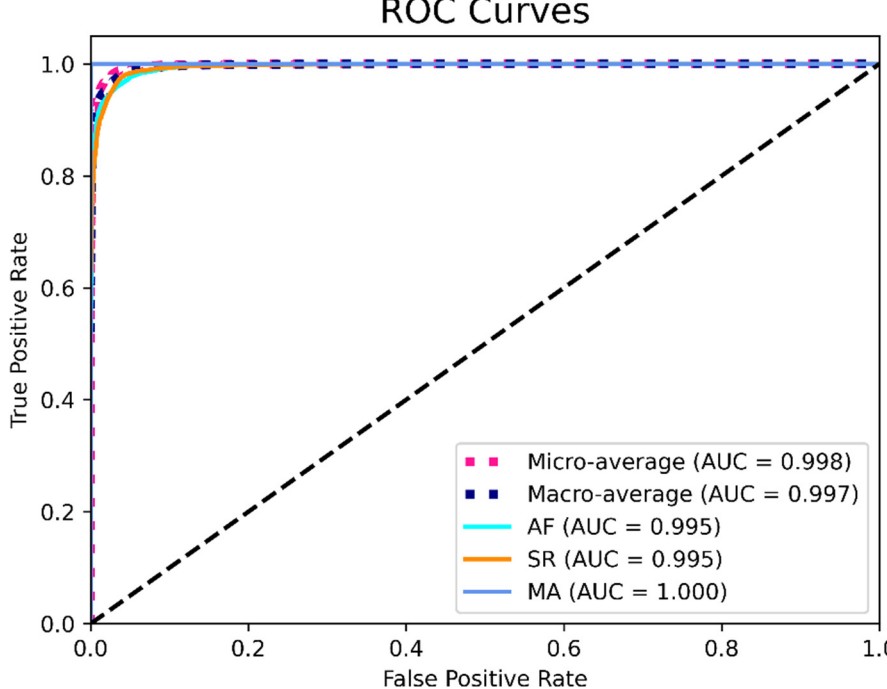

**Figure 7.** ROC curves based on the results from all 10 folds.

The data in Tables 6 and 7 indicate that MA could be identified without error, while AF and SR were sometimes misidentified. This result may have occurred because the MA

among subjects generally has much larger amplitudes than the two other classes (Figure 5), which tends to be easily identified due to strong feature extraction ability of the CNN model. The AF and SR signals are closer in amplitude, and the BCG signal itself has considerable individual variability, which contributes to the confusion between them. In addition, the sensitivity was observed to be lower than the specificity in AF, which indicates that AF is more likely to be misjudged than SR. This result may be due to the high variability of BCG signals. When AF occurs, complex and diverse internal forces cause the BCG signal to exhibit very irregular waveforms, which are difficult to interpret. At the same time, the rhythm of some SR BCG signals is not very regular (Figure 5), which makes AF more likely to be misjudged as SR. Table 7 shows that the model has good stability, with low standard deviations for all metrics.

Figure 7 indicates that the model achieved an AUC score of 0.995, 0.995, and 1.000 for AF, SR and MA, respectively, which shows good performance of the algorithm. To summarize, the final performance of the proposed residual CNN model achieved a mean precision, sensitivity, specificity, F1 score, and accuracy of 96.8%, 93.7%, 98.4%, 95.2%, and 96.8%, respectively.

## 4. Discussion

### 4.1. Method Analysis

In this study, we focused on developing a BCG-based method for household daily AF screening, which is mainly required by people with paroxysmal AF or with a high risk of AF. We first constructed a balanced BCG signal dataset containing AF, SR, and MA through data acquisition, preprocessing, labeling, and segmentation. The number of BCG segments in our dataset is larger than any other previous BCG-based study, ensuring enough BCG data for training the model. In previous studies [31–33], MA BCG signals were discarded during data preprocessing to focus on classification between AF and SR. We added the MA to dataset for two reasons. First, incorporating MA into the dataset allows the algorithm to handle the MA signals common in BCG recordings in real scenes, which improves the robustness of the algorithm. Second, the binary classification has the characteristics of either-or and three-class classification task is more challenging.

We applied the residual CNN to AF detection based on BCG segments for the first time in related studies. The proposed model has four modules: the downsampling convolutional module, local feature learning module, global feature learning module, and classification module. The downsampling convolutional module mainly undertakes two functions: to quickly reduce the dimension of the feature vector and to concentrate the contextual information of the BCG signal to prevent overfitting. The local and global feature learning modules are core components of the model. Stacked residual blocks are used for local feature extraction, and GlobalAvgPooling layer can capture global features from the feature vectors output by local feature learning module, which attempts more in-depth feature learning of BCG segment. We explored the impact of different numbers of residual blocks on the classification performance to determine the final network structure. Tenfold cross validation was applied to further confirm the stability of the model. The final confusion matrix and the values of various metrics show that our proposed method achieved good performance on the task of off-line BCG segment-based AF detection.

### 4.2. Method Comparison

We compared the proposed method with four previously published methods [30–33]. All four methods are based on the noncontact acquisition of BCG signals for AF detection. The methods published in [30–32] are based on traditional machine learning, which require artificial feature engineering. Jiang et al. [33] applied a DNN consisting of CNN and Bi-LSTM into BCG-AF detection. A detailed comparison is shown in Table 8.

**Table 8.** Method comparison.

| Methods | Brüser et al. [30] | Yu et al. [31] | Wen et al. [32] | Jiang et al. [33] | The Proposed |
|---|---|---|---|---|---|
| Segment length | 30 s | 30 s | 60 s | 24 s, 1 s | 10 s |
| Number of segments | 856 | 807 | 2915 | 2000 | **28,214** |
| Number of subjects | 10 | 12 | 37 | **59** | 45 |
| Number of classes | **3** | 2 | 2 | 2 | **3** |
| Pre | 90.7% | 94.6% | 92.8% | 93.7% | **96.8%** |
| Sen | 93.8% | **97.0%** | 96.8% | 95.9% | 93.7% |
| Spe | 98.2% | 89.1% | 92.0% | 93.5% | **98.4%** |
| F1 | 92.1% | - | 94.4% | - | **95.2%** |
| Acc | 92.1% | 94.4% | 94.5% | 94.7% | **96.8%** |

The data in Table 8 clearly show that the proposed method performs better than the previous methods in terms of the Pre, Spe, F1, and Acc, which indicates that our method is suitable for AF screening. In addition, we compared the segment length, number of segments, number of subjects, and number of classes to evaluate the feasibility and credibility of our method from multiple perspectives. Our BCG data were collected from 45 subjects and divided into 28,214 ten-second segments after preprocessing. The dataset contains a similar number of signals for all three signal types. The increased number of segments increases the size and diversity of the BCG dataset and the balanced data distribution improves the model training efficiency while ensuring that the generalizability of the classification algorithm is not reduced due to too much data from a certain class. In this manuscript, we completed a three-class classification task similar to Brüser et al. [30], which is more difficult than the two-class (only AF and SR) classification problem [31–33]. The addition of a motion artifacts class allows our method to handle motion artifacts that are common in BCG recordings, thus enabling unsupervised processing of the BCG signal [30]. The results show that the ability of our method to detect AF is not reduced by motion artifacts, which indicates that the method performs well in terms of anti-noise interference.

Compared with traditional machine learning algorithms, one advantage of the proposed method is that there is no need to manually extract features of BCG signals [30–32]. Compared with the deep learning method proposed by Jiang et al. [33], the proposed method has two distinct advantages. First, the number of segments is considerably larger, which is of great significance for training a deep learning model and improving the credibility of the method. Second, the subjects in [33] all suffered from paroxysmal AF, while the subjects in our research consisted of 19 people suffering from paroxysmal AF and 26 non-AF patients. Such an experimental setup enables MA BCG signals from both AF patients and non-AF individuals, increasing the diversity of the BCG dataset and making the AF detection experiments closer to real scenarios. The standard deviations of all metrics in Table 7 were less than 0.02, demonstrating that our method performs well in terms of preventing overfitting and improving generalizability.

### 4.3. Application Issues and Limitations

Compared with ECG, the advantage of BCG is that the signals can be acquired in a noncontact way, which ensures comfort and convenience of signal acquisition. However, unlike ECG, for which there are already well-defined diagnostic criteria, the use of BCG to detect arrhythmias such as AF is still in its infancy. BCG is accompanied by a high degree of variability and suffers from inadequate interpretation, as the relationship between BCG and cardiac activity has not been fully elucidated. Therefore, BCG may not replace ECG as the gold standard for clinical diagnosis at the current stage.

The proposed method in this manuscript provides a feasible solution for detect AF in BCG segments. Considering the easy acquisition of BCG, this method can be applied to AF screening, providing tools for timely detection of AF and follow-up after AF drug or device treatment. In clinical application, it may be used to determine whether AF occurs during

the sleep of the subject, and how many 10 s segments of AF have occurred. At the same time, it can also assist in determining whether these 10 s segments of AF are continuous, which all contribute to initial screening and subsequent more accurate clinical diagnosis and treatment of AF.

Furthermore, there are a few limitations to the proposed method. First, although we constructed a BCG dataset of 28,214 ten-second segments, which is greater in number than any other BCG study, and the number of subjects is considerable among related studies, the database still needs to be expanded to further evaluate the robustness of the deep learning algorithm. Second, the database mostly included older subjects. Third, arrhythmias other than AF were not considered.

### 4.4. Future Work

The purpose of our study is to apply an offline DNN model to identify AF segments based on BCG signals for long-term household AF screening. Considering practical application scenarios, we will continue to conduct research in the following aspects. First, due to the individual differences of BCG signals, we will attempt to collect more BCG signals of subjects of different ages to increase the size and diversity of the dataset. Different acquisition devices, acquisition conditions, and subject postures will also be taken into account to analyze performance of the method in different scenarios. Second, we will explore the structure and training method of the DNN model to accommodate the expansion of the dataset. Since ECG-based methods are relatively mature, DNNs that were widely used in the ECG detection of AF will be given priority. Third, other arrhythmias, which may cause false positives in AF discrimination, should be considered during classification if a big enough dataset can be built. Thus, we may expand the method to more classes of cardiac diseases in future research.

### 5. Conclusions

In this manuscript, we proposed a feasible method for AF detection by BCG signals of 10 s length. We collected BCG signals from 45 inpatients and constructed a large BCG dataset containing 28,214 ten-second nonoverlapping segments, including AF, SR, and MA. An offline residual CNN model was designed to classify the three types of BCG signals. The model has an end-to-end classification structure, with a downsampling convolutional module, a local feature learning module, a global feature learning module, and a classification module. The model achieved precision, sensitivity, specificity, F1 score, and accuracy of 96.8%, 93.7%, 98.4%, 95.2%, and 96.8%, respectively. The results suggest that the proposed method may be used as a tool for AF screening in long-term household cardiac monitoring devices.

**Author Contributions:** Conceptualization, Q.S. and Y.H.; methodology, Q.S. and Y.H.; software, Q.S.; investigation, Q.S. and Y.H.; resources, Q.S., B.Z., P.L. and T.L.; data curation, Q.S., B.Z., P.L. and T.L.; writing—original draft preparation, Q.S.; writing—review and editing, Q.S., Y.H. and X.W.; visualization, Q.S.; project administration, Q.S. and X.W.; funding acquisition, X.W. All authors have read and agreed to the published version of the manuscript.

**Funding:** This work received financial support from National Natural Science Foundation of China, grant no. 1171009 and 61801123; the Shanghai Municipal Science and Technology Major Project, grant no. 2017SHZDZX01 and 16441907900; the Shanghai Municipal Science and Economic and Informatization Commission Project, grant no. GYQJ-2018-2-05; Medical Engineering Fund of Fudan University yg2021-38; Zhejiang Provincial Natural Science Foundation of China, grant no. LY20H090001.

**Institutional Review Board Statement:** The study was conducted according to the guidelines of the Declaration of Helsinki, and approved by the Ethics Committee of Sir Run Run Shaw Hospital, School of Medicine, Zhejiang University. (Approval no. Research 20190520-67).

**Informed Consent Statement:** Informed consent was obtained from all subjects involved in this study.

**Data Availability Statement:** Data sharing not applicable.

**Acknowledgments:** We would like to thank Hangzhou BOBO Technology Ltd. for providing original BCG data for our research.

**Conflicts of Interest:** The authors declare no conflict of interest.

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
