# Peer review of "Atrial Fibrillation Detection Based on a Residual CNN Using BCG Signals"

_electronics, doi:10.3390/electronics11182974_

Round 1
Reviewer 1 Report (New Reviewer)
The authors developed an AF detection method by analyzing the measured BCG signals with a CNN-based architecture. Some major revisions are required.
1) Authors have given insufficient mention of previous AF detection studies. More related work is required.
2) The contributions and differences of the study should be clearly stated.
3) Training parameters of the CNN model should be shared (epoch number, mini-batch, etc.)
4) Knowledge of PC running deep learning is required.
5) Are there any datasets similar to the produced dataset? Or what are the differences?
6) The expression "Deep Residual" should be explained.
7) A discussion section should be created and future studies should be explained.
Author Response
Please see the attachment.

Reviewer 2 Report (New Reviewer)
An interesting effort was done by the authors of the paper "Atrial Fibrillation Detection Based on a Deep Residual CNN Using BCG Signals" in addressing a large challenge related to the detection of atrial fibrillation.
I would like to provide the following suggestions:
1) I would recommend using % when reporting CNN scores.
2) I would suggest add “years” or something like this after “ages of 65 and 85” in line 40
3) I would propose using a more appropriate term for "scholars" in line 73.
4) I would propose adding references in lines 82-84
5) I would suggest rephrase lines 85-87 to make the focus of your paper more congruent with an off-line CNN architecture to categorize AF.
6)Section 2.1 should, in my opinion, be refocused. Additionally, considering the height of an average mattress and how sensitive this device is to a typical person's movement without adding artifacts. I'm curious to know how sensitive it is to continuously monitoring the BCG signal.
7)What does N in Equation 1 indicate?
8) Has a threshold that is lower or higher been selected in accordance with the literature?
9) For processing and dataset organizing, I would recommend "A Machine Learning Approach Involving Functional Connectivity Features to Classify Rest-EEG Psychogenic Non-Epileptic Seizures from Healthy Controls."
10) Understanding the dataset structures, such as raws x columns, is essential to comprehending how data were fed into CNN.
11) Figure 5's plots lack axis information, which may be informatively included. Additionally, I would advise increasing the labels' font.
12) I would suggest improving the training set formula as in line 183 because it seems to not be working for multiclass.
13) It is well known that the CNN architecture consists of Conv, Relu, Conv, Relu, Conv, and Pool. Is your architecture specially designed after multiple tests utilizing your data?
14) ROC curves would provide a more visually appealing representation of classification challenges across classes.
15) Table 5 shows that while searching in MA, the model seems to be overfit. I would advise adequately addressing this important point in the results section of the manuscript.
16) In Table 7 I would suggest +/- STD
17) In line 300 I would advise using %
18) I would advise enhancing the results section.
Round 2
Reviewer 1 Report (New Reviewer)
The revisions are sufficient, the article is acceptable.
Reviewer 2 Report (New Reviewer)
During the review process, the authors of the paper titled "Atrial Fibrillation Detection Based on a Deep Residual CNN Using BCG Signals" did an excellent job of enhancing the manuscript as a whole.
This manuscript is a resubmission of an earlier submission. The following is a list of the peer review reports and author responses from that submission.
Round 1
Reviewer 1 Report
The authors present a neural network for AF detection using ballistocardiogram signals. The study shows interesting results and high accuracy to differentiate AF and sinus rhythm.
Comments:
- The introduction is extensive and the examples in line 74 – 86 should be moved to a discussion section, where they can be compared with the results by the author (as in Table 7).
- The data in Table 1 is incomplete. Some baseline demographic data would be useful, such as age, blood pressure, medication profiles. Further, did the 19 patients with AF have persistent AF or paroxysmal AF. Where all their recordings in AF?
- Where motion artefact segments completely artefact or a combination of SR/AF with short episodes of motion artefacts lasting only a few seconds?
- Why did the authors choose to use 10 second segments? Longer segments, as in the other studies presented in Table 7, could reduce the performance, for example by sections which contain both motion artefacts and normal rhythms.
- Including the motion artefacts when calculating the mean performance, does not truly represent the performance. As expected, the motion artefacts are easily detected due to the higher amplitude which does not contain any repeatability (figure 3). This reviewer would prefer to leave the results on motion artifacts in the manuscript, however, focus the discussion and mean performance statistics only on AF and SR.
- Due to the fact that all signals were obtained in only 45 patients, the performance of the model may be overestimated. Future studies should study the impact of physical conditions on the performance of the model.
- Overall, the authors should discuss how this can be used in clinical practice where we do not use 10 second BCG segments. For example, the clinical question could be whether a patient had any AF overnight. Currently, their methods do not answer this question but have a high potential to do so.
Reviewer 2 Report
The paper is about the detection of atrial arrythmia using the BCG, an avenue that is interesting to explore.
The authors fail to deliver the necessary technical details about the used BCG acquisition system, namely sensor location, and number of sensors. Without this information all the work is compromised.
The built BCG dataset is badly explained. The demographics of the dataset should be explained in a atable , along with acquisitions conditions. Average acquisition duration should be referred.
The authors should also refer more prolifically the advantages and the disadvantages of the BCG relatively to the ECG. Admittedly, motion artefacts such as subject respiration, are a major problem. How did the authors deal with this? The application of the RMS filter is presented in a few sentences, clearly this preprocessing step should be elaborated upon on. Was the RMS filter applied to all BCG´s or only BCG´s that contained motion artifacts? If this is the case where the motion artifacts visually identified? Why the need and aim of normalizing the output of the RMS filter. Is the RMS filter the only artifact removal (?) method applied in this work?
Clearly, the problem of the motion artifacts should be further worked.
Number of epochs and batch size referred in page 225 should be explained.
Reviewer 3 Report
- The novelty of proposed computational techniques is not clear.
- The manuscript lacks the clear and complete detail on the computational techniques and experiments.
- The exact values of parameters/variables need to be provided. The reasons on choosing such values also need to be discussed. Is this the set of parameters/variables providing the best performance?
- Furthermore, computational experiments need to be run multiple (randomly) times.
- The number of subjects is rather small.
- The demographics of subjects need to be provided.
Round 2
Reviewer 3 Report
The manuscript was significantly revised. However, there are some concerns remaining.
- The detail of data acquisition can be further improved.
- The detail of computational setup and experiments can be further improved.
- The total size of data (both included and excluded ones) needs to be clearly provided.
- The epoch with length of 10 second is considerably small for determining/diagnosing the cardia conditions (e.g., AF).
- In addition, some details are not provided. For example, the posture of sleeping should to be mentioned.
